# The Landscape of Genetic Variation and Disease Risk in Romania: A Single-Center Study of Autosomal Recessive Carrier Frequencies and Molecular Variants

**DOI:** 10.3390/ijms262210912

**Published:** 2025-11-11

**Authors:** Miruna Gug, Nicoleta Andreescu, Lavinia Caba, Tudor-Alexandru Popoiu, Ioana Mozos, Cristina Gug

**Affiliations:** 1Doctoral School of Medicine, “Victor Babes” University of Medicine and Pharmacy, 300041 Timisoara, Romania; miruna.gug@umft.ro (M.G.); tudor.popoiu@umft.ro (T.-A.P.); 2Department of Microscopic Morphology, Discipline of Genetics, Faculty of Medicine, “Victor Babes” University of Medicine and Pharmacy, 300041 Timisoara, Romania; cristina.gug@umft.ro; 3Medical Genetics Office Doctor Gug, Timisoara 300200, Romania; 4Genomic Medicine Centre, “Victor Babes” University of Medicine and Pharmacy, 300041 Timisoara, Romania; 5Department of Medical Genetics, Faculty of Medicine, “Grigore T. Popa” University of Medicine and Pharmacy, 700115 Iasi, Romania; 6Department of Functional Sciences, Medical Information and Biostatistics Discipline, Faculty of Medicine, “Victor Babes” University of Medicine and Pharmacy, 300041 Timisoara, Romania; 7Center for Translational Research and Systems Medicine, “Victor Babes” University of Medicine and Pharmacy, 300173 Timisoara, Romania; ioanamozos@umft.ro; 8Department of Functional Sciences-Pathophysiology, “Victor Babes” University of Medicine and Pharmacy, 300173 Timisoara, Romania

**Keywords:** carrier test, NGS, allele frequency, genetic heterogeneity

## Abstract

Autosomal recessive (AR) disorders represent a significant public health challenge, as asymptomatic carriers are often unaware of their reproductive risks. This study provides the first comprehensive assessment of AR gene variant frequencies and their molecular landscape in a fertile Western Romanian population. Genetic results from 604 unrelated, unaffected Caucasian individuals of reproductive age, tested at a single genetic center between 2020 and 2024, were retrospectively analyzed. Next-generation sequencing (NGS) with a multi-gene panel targeting 300 AR-associated genes was used for molecular profiling. Variants were identified in 156 genes, with 75% of individuals carrying at least one AR variant (mean 1.77 variants/person). A subgroup with >3 pathogenic variants comprised 7.5%, posing a notable risk for future offspring. The most frequent variants were detected in *HFE* (1:5), *CFTR* (1:9), *BTD* (1:16), *GJB2* (1:17), and *CYP21A2* (1:19). Four variants (*HFE*, c.187C>G; *BTD*, c.1330G>C; *CFTR*, c.1210-34TG[11]T[5]; *GALT*, c.-119_-116del) were particularly prevalent, each exceeding 3% frequency. Considerable allelic heterogeneity was observed for distrinctive variants in *CFTR* (14), *PAH* (12), *USH2A* (12), and *ATP7B* (9). Several variants were linked to severe disorders, with *CFTR*, *GALT*, *ATP7B*, and *SMN1* identified as “red zone” genes associated with high morbidity and mortality. Low-frequency variants formed a “long tail” (83.9%), reflecting marked population heterogeneity and potential hidden disease risks. The study reveals high allelic diversity and a strong prevalence of AR variants in Western Romania. Variant-based gene classification supports population-level screening, highlighting the public health value of a national program to identify carriers and prevent severe inherited disorders.

## 1. Introduction

Autosomal recessive (AR) disorders represent a major health concern due to their often-severe clinical consequences and their silent mode of inheritance. While carriers of pathogenic or likely pathogenic (PLP) variants are typically asymptomatic, they can transmit these alleles to their offspring, resulting in a 25% risk of having an affected child if both parents carry variants in the same gene. Carrier frequency estimates and variant distribution are therefore essential for accurate reproductive risk assessment and for the development of preventive strategies through genetic counseling and screening programs [1,2,3].

Early estimations of AR carrier burden relied on indirect approaches. It was initially predicted that each individual carries approximately eight heterozygous deleterious variants [2]. Studies of consanguineous populations later suggested a burden of 3–5 heterozygous pathogenic variants per individual [3]. In the genomic era, population-scale sequencing has revealed even greater allelic heterogeneity, with predictions of up to 100 PLPs per individual [4]. Data from founder populations such as the Hutterites estimated that each founder carried on average 0.58 lethal AR variants [5]. More recent large-scale screening studies reported a prevalence of 0.4 lethal AR pathogenic variants per person using targeted panels, while genome-wide extrapolations suggest substantially higher burdens [6]. These discrepancies highlight the importance of population-specific studies.

Despite significant advances in next-generation sequencing (NGS) and large population-based carrier screening initiatives, there are still many European regions with limited or no genetic epidemiology data. In Romania, particularly in the Western region, no comprehensive study has previously addressed the prevalence, spectrum, and distribution of AR variants in the general population. This lack of data hampers the implementation of evidence-based carrier screening strategies and limits accurate counseling for reproductive decision-making. Carrier genetic testing provides critical information for reproductive risk assessment and family planning. Although carriers are generally unaffected, they do have a 50% chance of transmitting each pathogenic heterozygous allele to their children. If both partners are carriers of a mutation in the same gene, there is a 25% risk that their children will inherit the condition. Generating regional data is therefore crucial for improving reproductive counseling and preventive healthcare.

Carrier genetic testing, based on NGS multi-gene panels, has become a valuable tool for assessing individual and couple-based reproductive risks. Such testing not only identifies frequent pathogenic variants in well-known genes but also uncovers a wide spectrum of allelic heterogeneity in less common conditions. Understanding these regional patterns of AR variant distribution is crucial, as founder effects, consanguinity rates, and population history can significantly influence local genetic landscapes [7,8].

The present study provides the first comprehensive investigation of AR carrier frequencies and molecular variants in a fertile Western Romanian population. By analyzing 300 AR-associated genes in 604 unrelated individuals, we aimed to: (i) determine the prevalence of clinically significant variants, (ii) identify genes and disorders associated with increased genetic risk, and (iii) classify the observed variants, based on their frequency, to inform implementation of carrier screening and genetic counseling strategies in Romania.

## 2. Results

This retrospective study was carried out over a five-year period (January 2020–December 2024) at a single Genetics Center in Western Romania. A total of 604 unrelated, unaffected Caucasian individuals were included, comprising both males (20–54 years old) and females (19–47 years old), representing a fertile cohort. Genetic analysis was conducted using the Invitae Comprehensive Carrier Panel, which includes 300 genes associated with autosomal recessive inheritance (Appendix A). Across this cohort, pathogenic and likely pathogenic (PLP) variants were detected in 156 genes with AR variants, while the remaining 144 genes showed no detectable variants.

The data were further evaluated from multiple perspectives, including: (i) gene-specific variant frequency, (ii) types of mutations identified, (iii) allelic heterogeneity, (iv) classification of major risks according to morbidity, and (v) classification of disease risks according to mortality.

### 2.1. Gene Frequency

Among the 604 individuals analyzed, 453 (75%) carried at least one autosomal recessive (AR) variant classified as pathogenic or likely pathogenic (PLP), whereas 151 individuals (25%) presented no detectable variants in any of the genes included in the panel. Among participants with positive findings, the number of identified variants ranged from one to six, with an average of 1.77 variants per individual. Most carriers, 238 of 604 (39.4%), while 35.6% of participants carried variants in two to six different genes (Figure 1). Notably, two individuals carried variants in six different genes, each mapped to a separate locus.

The most frequently observed autosomal recessive (AR) genes carrying PLP variants in our cohort (*HFE*, *CFTR*, *BTD*, *GJB2*, *CYP21A2*, *GALT*, *SERPINA1*, *PAH*, *SMN1*, *ATP7B*, *USH2A*, and *WNT10A*) showed frequency patterns that differed from those reported in the pan-ethnic dataset (Table 1).

We classified the genes according to their carrier frequency (Figure 2) as follows:High frequency (up to 1:50): 12 genes.Moderate frequency (between 1:51 and 1:100): 13 genes.Low frequency (between 1:101 and 1:150): 19 genes.Very low frequency (greater than 1:151): the remaining 112 genes.

The highest carrier frequencies were observed for the following 12 genes: *HFE* (1:5), followed by *CFTR* (1:9), *BTD* (1:16), *GJB2* (1:17), *CYP21A2* (1:19), *GALT* (1:19), *SERPINA1* (1:26), *PAH* (1:27), *SMN1* (1:30), *ATP7B* (1:36), *USH2A* (1:43), and *WNT10A* (1:46).

The second category, comprising 13 genes with moderate frequency, includes 5 genes (*ACADM*, *ALDOB*, *DHCR7*, *GAA*, *HBA1*) with a carrier frequency of 1:60, followed by 3 genes (*EVC*, *SLC26A2*, *TPP1*) with a frequency of 1:86, and 5 genes (*COL7A1*, *CYP11B2*, *GBA1*, *NEB*, *NR2E3*) with a frequency of 1:100.

The third category, consisting of 19 genes with low frequency, includes 6 genes (*ACAD9*, *BBS1*, *CAPN3*, *GALC*, *SLC12A3*, *SLC22A5*) with a carrier frequency of 1:121, as well as 13 genes (*ARSA*, *CPT2*, *CRB1*, *EYS*, *G6PD*, *GBE1*, *HEXA*, *LAMA2*, *LDLR*, *LIPA*, *MEFV*, *NPC1*, *VPS13B*) with carrier frequencies ranging from 1:101 to 1:150.

The fourth category, which represents the largest group, includes 112 genes with very low carrier frequency (>1:151). Although individually rare, together they represent 71.8% and constitute a substantial portion of the genetic heterogeneity observed in the cohort.

### 2.2. Type of Mutated Variants Identified

A total of 802 clinically significant PLP variants were identified in our cohort, corresponding to 326 unique variant types (Appendix A). Among the 326 distinct variants, missense variants represented the largest category (49.1%, *n* = 160), followed by nonsense variants (29.1%, *n* = 95), which included 54 frameshift (16.6%) and 46 stop-gain (14.1%) variants. Intronic variants accounted for 16.0% (*n* = 52), comprising splice donor (5.8%), site/non-coding (5.5%), and splice acceptor (4.6%) variants. Deletions and duplications together constituted 5.2% (*n* = 17), including exon deletions (2.8%), small deletions/duplications of 1–3 base pairs (1.5%), and whole-gene deletions (0.9%). A single RNA change (0.3%) was also observed.

When considering the total of 802 variants identified in our cohort, missense variants again predominated (56.7%, *n* = 455), followed by nonsense variants (19.7%, *n* = 158), which comprised 87 frameshift (10.8%) and 71 stop-gain (8.9%) events. Intronic variants accounted for 15.6% (*n* = 125), including splice donor (2.6%), site/non-coding (9.8%), and splice acceptor (3.1%) variants. Deletions and duplications represented 7.7% (*n* = 62), consisting of exon deletions (2.9%), small deletions/duplications (2.9%), and gene deletions (2%). Two RNA changes (0.2%) were also detected (Figure 3).

### 2.3. Gene Rankings

We further classified the genes according to the frequency of identified PLP variants. Out of 802 total variants recorded, 11 variants across 9 genes exhibited a frequency > 1%. Among these, one *HFE* variant exceeded a frequency of 12%, a *BTD* variant was higher than 4%, 2 variants (one in *CFTR* and one *GALT*) were >3%, while 3 variants (one in GJB2, one in *HFE*, and one in *CFTR*) reached a frequency > 2% (Table 2). The most frequently implicated genes were *HFE* (2 variants), *BTD*, *CFTR* (2 variants), *GALT*, *GJB2*, *SMN1*, *WNT10A*, and *CYP21A2*.

### 2.4. Allelic Heterogeneity

In total, we identified 802 variants in 469 individuals who carried at least one clinically significant variant, corresponding to an average of 1.77 variants per individual. A high degree of allelic heterogeneity was observed in several genes. The most notable examples included *CFTR* (13 distinct variants in 66 carriers), *PAH* (12 variants in 21 carriers), *USH2A* (12 variants in 14 carriers), *ATP7B* (9 variants in 17 carriers), and *CYP21A2* (7 variants in 29 carriers) (Table 3).

Most individuals carried a single heterozygous variant per gene. Exceptions were observed in a few cases: two individuals exhibited double heterozygosity across two genes (*CFTR* and *HFE*), and one individual carried six variants in the *CYP21A2* gene. These instances, involving biallelic pathogenic variants, increase the likelihood of a clinical phenotype and highlight the importance of careful variant interpretation.

### 2.5. Classification of Diseases Risk by Morbidity and Mortality

In the analyzed cohort, autosomal recessive disease–associated genes showed variable morbidity and mortality, ranging from mild to life-threatening outcomes (Table 4).

The high-frequency group (12 genes) was mainly linked to metabolic, neuromuscular, and sensory disorders, several of which (*CFTR*, *GALT*, *PAH*, *SMN1*, *ATP7B*) were associated with severe or early-onset forms and high mortality if untreated. The moderate-frequency group (13 genes) included disorders with high morbidity and frequent infantile lethality, while the low-frequency group (19 genes) encompassed diverse metabolic, neuromuscular, and neurodegenerative conditions of variable severity.

Overall, the data illustrate the heterogeneity of genetic disease burden in the Western Romanian population. The actual risk for offspring depends on the carrier status of both reproductive partners, emphasizing the importance of comprehensive carrier screening in identifying couples at increased risk for severe autosomal recessive diseases.

## 3. Discussion

This study presents the first comprehensive characterization of autosomal recessive carrier variants across 300 genes in a Romanian cohort, providing valuable population-level genomic data from Eastern Europe (Appendix A). In contrast to previous research that focused mainly on affected individuals or newborns, our analysis of a healthy, reproductive-age population offers a more accurate depiction of carrier frequencies within the general population. These findings address a major regional knowledge gap and capture the diverse, multiethnic, and historically rich background of Western Romania.

Participants carried on average 1.77 pathogenic or likely pathogenic (PLP) variants (Figure 1), consistent with carrier loads described in European populations [1]. Such distribution patterns illustrate the genetic heterogeneity typical of mixed populations and the cumulative impact of multiple low-frequency variants. Our findings also emphasize that the regional genetic landscape has been shaped by founder effects, migration, and historical admixture, factors that collectively influence variant prevalence and distribution. 

Among the 802 identified pathogenic or likely pathogenic (PLP) variants, 56.7% were missense variants, 19.7% were nonsense variants, approximately 15.6% were intronic variants, and deletions and duplications collectively accounted for 7.7% of all variants (Figure 3).

We identified eight genes—*HFE* (two variants), *BTD*, *CFTR* (two variants), *GALT*, *GJB2*, *SMN1*, *WNT10A*, and *CYP21A2*—comprising ten variants whose individual frequencies exceeded 1% in our cohort (Table 2). Among these, four variants (*HFE:* c.187C>G, *BTD:* c.1330G>C, *CFTR:* c.1210-34TG[11]T[5], *GALT:* c.-119_-116del) were particularly prevalent, each exceeding a 3% frequency. Notably, *CFTR* and *CYP21A2* exhibited both high allelic heterogeneity and variant frequencies above 1%, emphasizing their clinical relevance in carrier screening within the Romanian population.

The genes with the highest carrier frequencies (>1:50)—*HFE*, *CFTR*, *BTD*, *GJB2*, *CYP21A2*, *GALT*, *SERPINA1*, *PAH*, *SMN1*, *ATP7B*, *USH2A*, and *WNT10A*—represent that represent the bulk of reproductive risk in our population. (Table 1). Many of these disorders are severe or life-limiting if untreated and align with ACMG and ACOG recommendations for inclusion in reproductive carrier screening [9,10]. The presence of multiple founder mutations (e.g., *CFTR*: p.Phe508del, *GJB2*: c.35delG) further highlights how population history and migration shaped these elevated carrier rates in Europe [11,12].

The high frequencies of these alleles in our population underline their potential for integration into national or regional screening programs. Notably, the pattern observed in Western Romania mirrors trends in other European cohorts [1,6], confirming the importance of local genetic mapping to refine population-specific screening strategies. 

Several of these genes identified are of critical clinical importance, being associated with early-life morbidity and mortality. Variants in *CFTR*, *GALT*, *ATP7B*, and *SMN1* fall within the so-called **“red zone,”** where the absence of timely diagnosis and treatment can lead to lethal neonatal or childhood outcomes (Table 4).

In Romania, the national neonatal metabolic screening program currently covers three disorders, corresponding to *CFTR*, *PAH* and hypothyroidism, together with clinical screening for deafness [13,14]. In the private healthcare sector, expanded panels—screening up to 50–100 metabolic diseases—are available on request. Early diagnosis is undoubtedly beneficial but often associated with significant psychological stress for families. By contrast, knowledge of parental carrier status can provide reassurance when no risks are identified or enable timely management through planned prenatal testing when risks are present.

Building on our results, where 156 of the 300 genes tested harbored PLP variants, we sought to better understand the distribution of carrier burden across the population. By classifying genes according to carrier frequency, we can distinguish those that pose the greatest reproductive risk from those that contribute mainly to genetic heterogeneity. When considering the distribution of carrier frequencies across the four categories, some important patterns emerge.

Metabolic disorders dominate the cohort: six high-frequency genes (*HFE*, *CFTR*, *ATP7B*, *GALT*, *PAH*, *BTD*), five medium-frequency genes (*ACADM*, *ALDOB*, *DHCR7*, *GAA*, *GBA1*), and six low-frequency genes (*ACAD9*, *SLC12A3*, *SLC22A5*, *GBE1*, *LDLR*, *LIPA*), collectively associated with 17 important genetic diseases for Western Romania (Table 4). Carrier screening is recognized as a primary prevention strategy for inherited metabolic disorders [12,15].

In accordance with the recommendations of the American College of Medical Genetics and Genomics (ACMG) and the American College of Obstetricians and Gynecologists (ACOG), which prioritize carrier screening for severe early-onset disorders that cause cognitive impairment, require medical or surgical intervention, impact quality of life, and have an expected carrier frequency >1:100 [9,10], our analysis focused on this subgroup. Specifically, 25 high-priority genes were examined—12 with carrier frequencies up to 1:50 and 13 with frequencies between 1:51 and 1:100 (Table 1, Figure 2).

Hereditary hemochromatosis type 1 was the most prevalent disease-associated condition in the cohort, with variants in the *HFE* gene occurring at a frequency of 1:5. Within the *HFE* gene, the two well-characterized pathogenic variants exhibited markedly different frequencies, with H63D occurring approximately five times more frequently than C282Y. All individuals carrying high-risk genotypes (homozygous or compound heterozygote) were referred for hematological and gastroenterological evaluation, given the established association with iron overload, hepatic cirrhosis, and pancreatic insufficiency. Carrier frequency for *HFE* varies across populations: 1:3 in Northern Europe, 1:4 in Hispanics and pan-ethnic cohorts, and 1:5 in our Western Romanian cohort—slightly lower, yet clinically important given the morbidity of hemochromatosis. The C282Y variant is frequent in Northern Europe [16], while H63D is considered a European haplotype but has also been reported in both the United States and India [17]. 

*CFTR*-related variants were the next most common. The exonic variant F508del, originating as a founder mutation, accounted for 28.79% of carriers, substantially lower than the ~80% reported in North-Western Europe [18,19], but consistent with Central and Eastern European data. The detection of both exonic and intronic variants, many absent from standard ACMG-25 or *CFTR*-100 panels [7], reinforces the need to adapt test designs to regional variant spectra. This observation parallels findings in other genetically diverse countries, such as Turkey and France, where high *CFTR* heterogeneity complicates panel standardization [20]. Globally, *CFTR* is among the most heterogeneous human genes, with over 2100 variants reported [21]. Genotype–phenotype correlations reinforce the clinical relevance of these findings. Severe variants are typically associated with multi-organ CF, whereas mild variants predispose to monosymptomatic or adult-onset conditions, such as male infertility, bronchiectasis, or recurrent pancreatitis [21].

*BTD* gene, associated with biotinidase deficiency, showed a high carrier frequency (1:16) in our cohort, compared to Non-Finnish European (NFE) population (1:25). Nearly all carriers harbored the c.1330G>C (D444H) variant, suggesting a long-standing European founder effect, only one individual had a different allele (T532M). The D444H variant has previously been reported in several European countries and the United States [22], whereas T532M was identified in Turkey [23,24].

For *GJB2* gene, associated with *GJB2*-related conditions, the carrier frequency was 1:17, substantially higher than the NFE estimate of 1:42. Seven distinct variants were, with the frameshift mutation c.35del accounting for 60% of cases. This unexpectedly high prevalence suggests a markedly increased risk of hereditary deafness in the population of our region. The c.35delG mutation is well recognized as the most common cause of nonsyndromic hearing loss in populations of Caucasian origin. This variant, believed to have originated from a common ancestor (founder) in the Middle East or Mediterranean region and spread during Neolithic migrations [11], exemplifies the impact of ancient demographic processes on present-day carrier profiles.

*GALT* gene displayed a carrier frequency of 1:19 identical to that of the NFE population, with three variants identified in 32 individuals. The vast majority (78.13%) carried the non-coding intronic variant c.-119_-116del, which, when found in trans with a pathogenic allele, results in Duarte galactosemia. This mild form of galactosemia allows individuals to tolerate higher galactose levels compared with classic galactosemia. According to a recent study, based on clinical observations, it was established that carriers of Duarte galactosemia can follow unrestricted diets [25].

*CYP21A2* gene, associated with congenital adrenal hyperplasia due to 21-hydroxylase deficiency, showed a lower carrier frequency of 1:19, compared with a NFE population estimate of 1:17. Eight distinct variants were detected, the most frequent being the missense c.1360C>T, present in 32.25% and previously associated with virilization in reported in Turkish patients [26].

For *SERPINA1* gene, associated with alpha-1 antitrypsin deficiency, we detected carriers at a frequency of 1:26, lower than the NFE population frequency (1:18). Among the five variants, the most frequent was c.863A>T, a pathogenic missense variant identified in 30.43%, also reported in French-Canadian populations [27].

*PAH* gene, associated with phenylalanine hydroxylase deficiency, was observed with a frequency of 1:27. The most frequent variant in our country and globally was c.1222C>T (22.72%) which causes a severe increase in serum phenylalanine and does not comply with Phe monitoring guidelines. Given that PAH deficiency causes phenylketonuria (PKU), which is included in the Romanian neonatal screening program, these findings have direct clinical relevance.

*SMN1* gene, associated with spinal muscular atrophy was identified with a carrier frequency of approximately 1:30, higher than in the Caucasian populations. PLP variants in *SMN1* gene are associated with spinal muscular atrophy (SMA), recognized as the leading genetic cause of infant mortality [28]. Given the availability of effective gene therapies, identifying SMA carriers before conception can substantially improve reproductive counseling outcomes.

*ATP7B* gene, associated with Wilson disease, had a carrier frequency of in our cohort of 1:36, markedly higher than the NFE population rate of 1:50. Disease prevalence varies across populations, being highest in Asians [15] and Ashkenazi Jews, but lower in the UK and France [29]. In our cohort, we identified several PLP variants, of which the most frequent were two missense alleles (c.2817G>T and c.3207C>A) together accounting for 47.05% of cases. The c.3207C>A variant is a founder variant that was identified in the Roma population of Bulgaria, Romania, Hungary, Germany, and France [30].

*USH2A* gene was detected in carriers with a frequency of 1:43, higher than the NFE population frequency of 1:70 and is represented by 12 variants. *USH2A*-related conditions are associated with the clinical picture characterized by vision and hearing loss from birth.

In the Western Romanian population studied, the risk of deafness is given by pathogenic variants in 7 genes included in the panel: *GJB2*, *USH2A*, *LOXHD1*, *SLC26A4*, *USH1C*, *COL4A4*, *COL4A3*, present in almost 10% of the cohort, therefore representing a significant risk. 

Carriers of the *WNT10A* gene, causing *WNT10A*-related conditions, were identified at a frequency of 1:46, lower than the reported NFE frequency of 1:33. All cases were represented exclusively by the missense variant c.682T>A (p.Phe228Ile), indicating a possible founder effect. This variant has previously been described in the Italian population in association with ectodermal dysplasia–related phenotypes, particularly oligodontia and hypodontia [31]. 

The recent publication of a Catalogue of autosomal recessive inherited disorders found among the Roma population in Europe [32] with the specification of the founder effect, allowed us to make a comparison with our data. We identified in western Romania the following genes and variants: *ACADM* (c.985A>G), *ATP7B* (c.3207C>A), *CFTR* (c.1624G>T), *CYP21A2* (c.293-13A/C>G), *SLC12A3* (c.1180+1G>T), *SLC22A5* (c.844del*) of which only the first two have been reported from Romania, the rest being present in the Roma population, spread in different European countries [32].

The low-frequency (1:101–1:150) and very low-frequency (>1:151) groups, although individually rare, collectively account for the largest fraction of allelic heterogeneity. In our study, this latter group is the largest, representing 83.9% genes with PLP variants. This pattern reflects the “long tail” effect described in recent genomic studies, where the cumulative impact of rare variants contributes significantly to disease risk [12]. This highlights the need for broad genomic approaches, since panels restricted to common variants would miss a substantial number of clinically relevant findings.

From a public health perspective, these findings have several implications. First, genes with high carrier frequencies (≤1:50) should be prioritized for targeted awareness campaigns and inclusion in reproductive carrier panels. Second, moderate-frequency genes (1:51–1:100), particularly those associated with treatable metabolic or neuromuscular disorders [33], are strong candidates for integration into national newborn screening programs. Finally, rare variants emphasize the importance of broad genomic testing in clinical and reproductive medicine [7,34]. Together, these insights support the creation of tiered screening strategies tailored to the Romanian population.

The identification of multiple carriers for variants known to cause severe or lethal diseases—such as those in *CFTR*, *GALT*, *SMN1*, and *ATP7B*—underscores the critical importance of early detection. Timely carrier identification can guide reproductive decisions and prevent neonatal morbidity and mortality. While prenatal and preimplantation genetic diagnosis [34,35] remain effective preventive tools, preconception screening offers greater flexibility and lower psychosocial burden by allowing couples to consider reproductive options before pregnancy.

Romania’s national genetic screening currently remains limited in scope and accessibility. Broader implementation of carrier testing is constrained by costs, low public awareness, and limited access to high-quality genetic counseling [36]. Nevertheless, the high prevalence of actionable variants identified here provides strong justification for expanding screening efforts. Integrating carrier testing into routine reproductive health assessments would align national practices with international recommendations and significantly improve preventive healthcare outcomes.

At a broader scale, population genomics offers essential insight into disease architecture and evolution. Understanding how population history, demographic dynamics, selection pressures, and founder effects have shaped allele distributions in Eastern Europe helps interpret clinical data and supports the design of equitable, population-specific genetic testing programs [37]. Mapping hotspot recurrent pathogenic variants also facilitates differentiation between inherited and de novo mutations, improving diagnostic accuracy. As genomic data from diverse regions of the world continue to accumulate, these resources can be made accessible to researchers through publications or online databases [38].

Carrier screening thus serves not only as a reproductive health measure but also as a cornerstone of preventive genomic medicine. By clarifying the distribution of pathogenic variants in the general population, it provides a framework for risk assessment, counseling, and informed decision-making [35,36]. Implementation of such programs—supported by robust databases and public education—has the potential to reduce the burden of recessive diseases and improve genetic education in Romania.

## 4. Materials and Methods

### 4.1. Selection of Participants and Clinical Data

A retrospective study was conducted on data obtained from 604 unaffected, unrelated Caucasian individuals (274 males and 330 females; male-to-female ratio 1:1.2) of reproductive age, ranging from 20 to 54 years in men and 19 to 47 years in women. All participants were tested between 1 January 2020 and 30 December 2024 at a single private genetic center located in Western Romania.

The genetic testing panel included 302 genes, of which 300 are associated with autosomal recessive (AR) inheritance, and two genes (F5 and F2) are linked to autosomal dominant (AD) inheritance. The present analysis focused exclusively on pathogenic and likely pathogenic (PLP) variants identified in AR genes; therefore, findings related to F2 and F5 were excluded. Although not included in the main analysis, 18 individuals carried PLP variants in F2 and 33 in F5. These AD gene variants are presented in a separate section of the same table, adjacent to the section listing AR genes with PLP variants identified in this study (Appendix A). For comparative purposes, carrier frequencies observed in our cohort were evaluated against the combined allele frequencies reported in gnomAD for the Caucasian population referred to as Non-Finnish European completed with Hardy-Weinberg law [39].

Variants classified as pseudodeficiency alleles—considered benign—were excluded from the analysis. The presence of a pseudodeficiency allele does not increase an individual’s carrier risk. These alleles may appear in test results because they can cause false-positive findings in certain biochemical assays, including newborn screening. However, pseudodeficiency alleles are not associated with disease causation, and carrier testing of the reproductive partner is not indicated for such variants. A list of the genes with pseudodeficiency alleles are provided in (Appendix A).

### 4.2. Ethical Approval

The study was conducted in accordance with the Declaration of Helsinki. Ethical approval was obtained from the Ethics Committee of Scientific Research at “Victor Babeș” University of Medicine and Pharmacy, Timișoara, Romania (No. 35, 3 June 2024). Written informed consent was obtained from all participants for genetic analysis, carrier screening, and the future publication of results.

### 4.3. Genetic Counseling

An experienced geneticist provided counseling to all participants in two sessions. During pre-test counseling, reproductive history, age, and any known or manifested hereditary conditions in the family were recorded, and family pedigrees were constructed. After receiving carrier screening results, a second counseling session was conducted to communicate any identified reproductive risks and their implications. For high-risk cases, prenatal genetic diagnosis via amniocentesis was recommended in future pregnancies, followed by targeted verification of the identified parental mutations.

### 4.4. Carrier Screen Test and Panel Genes

The analysis was performed on blood samples sent to Invitae Corporation (1400 16th Street, San Francisco, CA, USA, 94103). The Invitae Comprehensive Carrier Screen assessed 302 genes (Appendix A) for clinically significant pathogenic or likely pathogenic (PLP) variants associated with inherited conditions. Genomic DNA from each sample was enriched for targeted regions using a hybridization-based protocol and sequenced on an Illumina platform. All targeted regions were sequenced at a minimum depth of 50×, with additional analyses performed where necessary.

Sequencing reads were aligned to the reference genome (GRCh37), and sequence changes were interpreted within the context of a single clinically relevant transcript. Analysis focused on coding sequences, 10 bp of flanking intronic regions, and other genomic regions known to be disease-causing at the time of assay design. Promoters, untranslated regions, and other non-coding regions were not interrogated.

Exonic deletions and duplications were identified using an in-house algorithm that compares read depth for each target with the mean read depth and distribution from a set of clinical samples. Reportable variants were confirmed based on stringent criteria established by Invitae, using validated orthogonal approaches as needed [40]. Only variants with established clinical significance for the tested conditions were reported; variants of uncertain significance, benign variants, and likely benign variants were excluded.

### 4.5. Statistical Analysis of Variant Frequencies

All statistical analyses were performed using JASP and Excel Softwares. Descriptive statistics were used to summarize the distribution of variants and carrier status. The number of clinically significant genes per individual was calculated, and frequencies were reported as absolute counts and percentages. Genes were further classified by cohort frequency into high (1:1–1:50), medium (1:51–1:100), low (1:101–1:150), and very low (≥1:151) categories.

PLP variant types were grouped into nonsense, missense, intronic, and deletions/duplications, and subcategories were analyzed separately. The primary continuous variable (number of mutations/person) was the number of affected genes per individual. Descriptive statistics included mean, standard deviation (SD), median, mode, and range. Normality of distribution was assessed using the Shapiro–Wilk test, which indicated deviation from a normal distribution (*p* < 0.001) (Figure 4a). Consequently, results are reported as both mean ± SD and median with interquartile range (IQR) (Figure 4b).

## 5. Conclusions and Future Perspectives

This study represents the first comprehensive assessment of autosomal recessive (AR) carrier status within the Romanian population, identifying an average of 1.77 pathogenic variants per individual in the West region. Notably elevated carrier frequencies were observed for HFE, CFTR, BTD, GJB2, GALT, CYP21A2, SERPINA1, PAH, SMN1, ATP7B, USH2A, and WNT10A, with the greatest allelic heterogeneity detected in CFTR, PAH, USH2A, ATP7B, and CYP21A2. The most prevalent condition in this cohort was hereditary hemochromatosis, followed by cystic fibrosis, biotinidase deficiency, GJB2-related deafness, galactosemia, congenital adrenal hyperplasia, α1-antitrypsin deficiency, phenylalanine hydroxylase deficiency, spinal muscular atrophy, Wilson disease, USH2A-related disorders, and WNT10A-related conditions.

Collectively, these results define the principal genetic risks for AR disorders in Western Romania and highlight the importance of implementing carrier screening as part of preconception and prenatal care. Such integration can improve reproductive risk evaluation, guide family planning, and contribute to better clinical outcomes. Furthermore, the observed interpopulation variability in carrier frequencies underscores the relevance of these data as a resource for optimizing newborn screening programs at both regional and national levels. Ultimately, this study establishes a foundational reference for genetic counseling, as well as prenatal and preimplantation genetic diagnosis in Western Romania.

While these findings offer valuable insights into disease risk within the Romanian population, future large-scale studies employing carrier whole-exome sequencing (Carrier-WES) are expected to refine and expand our understanding of the genetic landscape and its implications for public health.

## Figures and Tables

**Figure 1 ijms-26-10912-f001:**
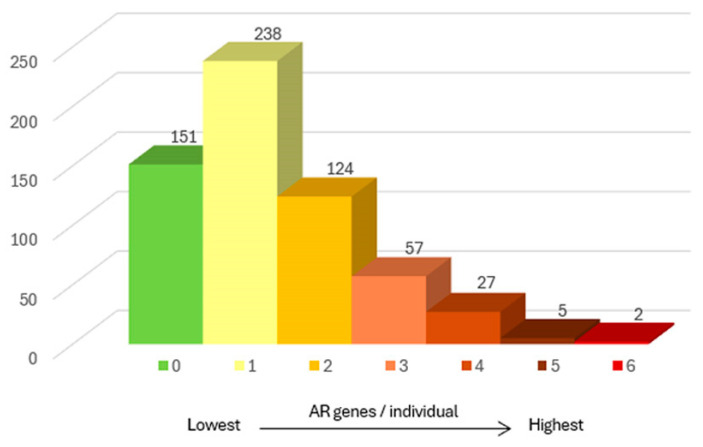
Number of clinically significant carrier genes among the individuals in the study.

**Figure 2 ijms-26-10912-f002:**
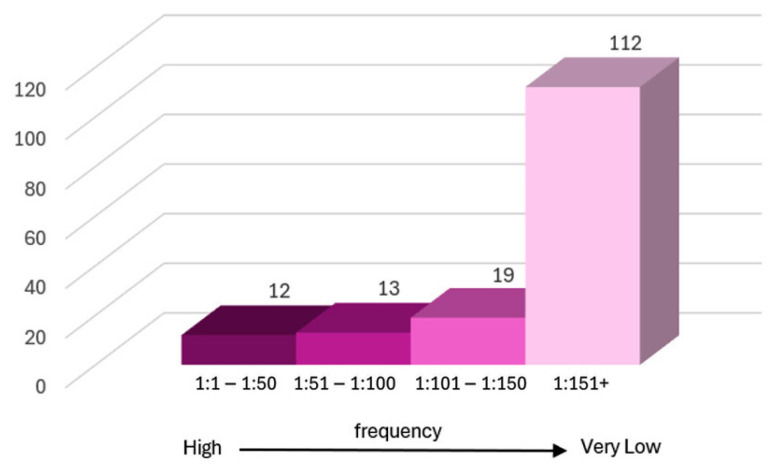
Gene groups classified by cohort frequency.

**Figure 3 ijms-26-10912-f003:**
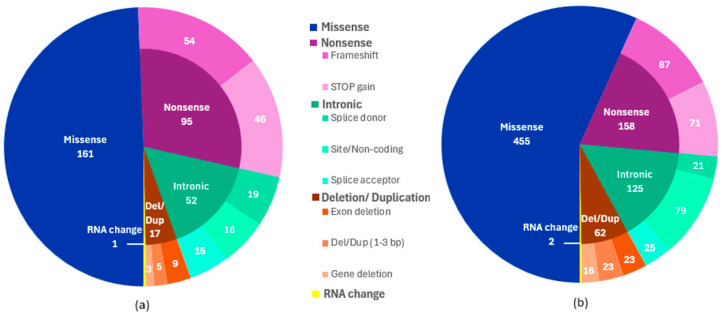
Distribution of variant types identified: (**a**) 326 unique variant types; (**b**) 802 unique variant types grouped by the number of individuals carrying each variant in the cohort.

**Figure 4 ijms-26-10912-f004:**
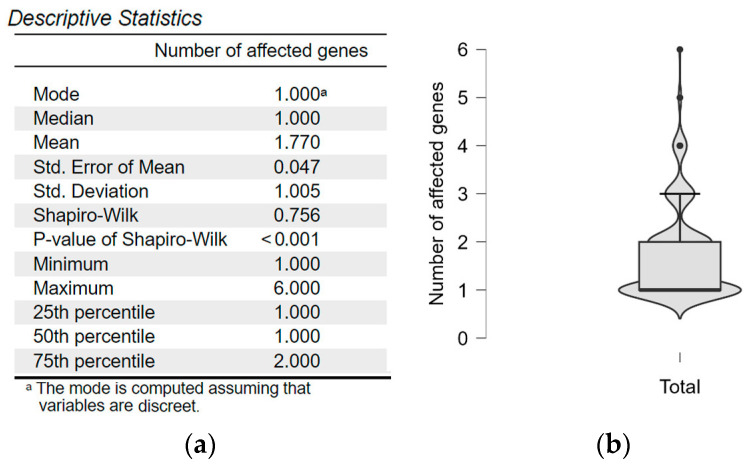
(**a**) Descriptive statistics; (**b**) Distribution of the number of affected genes.

**Table 1 ijms-26-10912-t001:** The most frequent autosomal recessive genes with PLP variants identified in our cohort, presented in comparison with the Caucasian population referred to as Non-Finnish European, according to the combined allele frequencies reported in gnomAD.

Disorder	Nomenclature	Inheritance	Gene	West RO ^1^ Carrier Frequency	Non-Finnish European Carrier Frequency
Hereditary hemochromatosis type 1	NM ^2^_000410.3	AR ^3^	*HFE*	1:5	1:6
CFTR-related conditions	NM_000492.3	AR	*CFTR*	1:9	1:9
Biotinidase deficiency	NM_000060.3	AR	*BTD*	1:16	1: 25
GJB2-related conditions	NM_004004.5	AR	*GJB2*	1:17	1: 42
Galactosemia (GALT-related)	NM_000155.3	AR	*GALT*	1:19	1:19
Congenital adrenal hyperplasia due to 21-hydroxylase deficiency	NM_000500.7	AR	*CYP21A2*	1:19	1:17
Alpha-1 antitrypsin deficiency	NM_000295.4	AR	*SERPINA1*	1:26	1: 18
Phenylalanine hydroxylase deficiency	NM_000277.1	AR	*PAH*	1:27	1: 50
Spinal muscular atrophy	NM_000344.3	AR	*SMN1*	1:30	1: 45
Wilson disease (AR)	NM_000053.3	AR	*ATP7B*	1:36	1:50
USH2A-related conditions	NM_206933.2	AR	*USH2A*	1:43	1: 70
WNT10A-related conditions	NM_025216.2	AR	*WNT10A*	1:46	1:33

^1^ RO = Romanian population, ^2^ NM = nomenclature, ^3^ AR = autosomal recessive.

**Table 2 ijms-26-10912-t002:** Genes with PLP variants exceeding 1% frequency in the cohort.

Gene	Variants ^1^	Number of Individuals	Variant Frequency in the Cohort (%)
*HFE*	c.187C>G (p.His63Asp) (H63D)	100	12.47
*BTD*	c.1330G>C (p.Asp444His)	36	4.48
*CFTR*	c.1210-34TG[11]T[5] (Intronic)	27	3.37
*GALT*	c.-119_-116del (intronic)	25	3.12
*GJB2*	c.35del (p.Gly12Valfs*2)	21	2.62
*HFE*	c.845G>A (p.Cys282Tyr) § ^2^ = C282Y	20	2.49
*CFTR*	c.1521_1523del (p.Phe508del) (F508del)	19	2.37
*SMN1*	Exon 7 + 8 deletion	14	1.75
*WNT1OA*	c.682T>A (p.Phe228Ile)	13	1.62
*CYP21A2*	c.1360C>T (p.Pro454Ser)	10	1.24
*HBA1*	Deletion (Entire coding sequence)	9	1.12

^1^ Variants are presented in order of frequency, ^2^ § = late penetrance.

**Table 3 ijms-26-10912-t003:** Genes with the highest allelic heterogeneity and their variants.

Gene	Variant ^1^	Legacy Name	Location	Number of Individuals
*CFTR*	c.1210-34TG[11]T[5]	-	Intronic	27
	c.1521_1523del (p.Phe508del)	F508del	Exonic	19
	c.1408A>G (p.Met470Val)	M470V	Exonic	5
	c.1210-34TG[12]T[5]	-	Intronic	3
	c.1807G>A (p.Val603Ile)	V603I	Exonic	2
	c.3472 C>G (Arg1158*)	R1158X	Exonic	2
	c.3909C>G (p.Asn1303Lys)	N1303K	Exonic	2
	c.377G>A (p.Gly126Asp)	G126D	Exonic	1
	c.1210-7_1210-6del	-	Intronic	1
	c.1210-11delinsGTG	-	Intronic	1
	c.1624G>T (p.Gly542*)	G542X	Exonic	1
	c.2813T>G (p.Val938Gly)	V938G	Exonic	1
	c.3846G>A (p.Trp1282*)	W1282X	Exonic	1
*PAH*	c.1222C>T (p.Arg408Trp)	R408W	Exonic	5
	c.898G>T (p.Ala300Ser)	A300S	Exonic	4
	c.143T>C (p.Leu48Ser)	L48S	Exonic	2
	c.673C>A (p.Pro225Thr)	P225T	Exonic	2
	c.529G>C (p.Val177Leu)	V177L	Exonic	1
	c.533A>G (p.Glu178Gly)	E178G	Exonic	1
	c.545A>G (p.Glu182Gly)	E182G	Exonic	1
	c.734T>C (p.Val245Ala)	V245A	Exonic	1
	c.844G>T (p.Val282Leu)	V282L	Exonic	1
	c.1066-11G>A	-	Intronic	1
	c.1208C>T (p.Ala403Val)	A403V	Exonic	1
	c.1315 + 1G>A	-	Intronic	1
*USH2A*	c.11864G>A (p.Trp3955*)	W3955*	Exonic	2
	c.12332C>T (p.Ser4111Phe)	S4111F	Exonic	2
	c.2296T>C (p.Cys766Arg)	C766R	Exonic	1
	c.2802T>G (p.Cys934Trp)	C934W	Exonic	1
	c.6937G>T (p.Gly2313Cys)	G2313C	Exonic	1
	c.7524del (p.Arg2509Glyfs*19)	R2509G	Exonic	1
	c.8618T>G (p.Leu2873*)	L2873*	Exonic	1
	c.8682-9A>G	-	Intronic	1
	c.10073G>A (p.Cys3358Tyr)	C3358Y	Exonic	1
	c.12268C>A (p.Pro4090Thr)	P4090T	Exonic	1
	c.12569T>C (p.Val4190Ala)	V4190A	Exonic	1
	c.14803C>T (p.Arg4935*)	R4935*	Exonic	1
*ATP7B*	c.2817G>T (p.Trp939Cys)	W939C	Exonic	4
	c.3207C>A (p.His1069Gln)	H1069Q	Exonic	4
	c.19_20del (p.Gln7Aspfs*14)	Q7D	Exonic	2
	c.347T>C (p.Ile116Thr)	I116T	Exonic	2
	c.1877G>C (p.Gly626Ala)	G626A	Exonic	1
	c.2305A>G (p.Met769Val)	M769V	Exonic	1
	c.2532delA (p.Val845Serfs*28)	V845S	Exonic	1
	c.2605G>A (p.Gly869Arg)	G869R	Exonic	1
	c.2906G>A (p.Arg969Gln)	R969Q	Exonic	1
*CYP21A2*	c.1360C>T (p.Pro454Ser)	P454S	Exonic	10
	c.844G>T (p.Val282Leu)	V282L	Exonic	6
	c.955C>T (p.Gln319*)	Q319*	Exonic	5
	c.293-13C>G	-	Intronic	4
	c.332_339del (p.Gly111Valfs*21)	G111V	Exonic	2
	c.188A>T (p.His63Leu)	H63L	Exonic	1
	c.1069C>T (p.Arg357Trp)	R357W	Exonic	1

^1^ Variants are presented in order of frequency.

**Table 4 ijms-26-10912-t004:** Classification of the main genetic disease risks, presented in descending order of pathological gene frequency observed in the studied cohort.

Gene Frequency in Western Romania	Gene	Physiological System/Type	Disease	Observations Related to Morbidity and Mortailty
High frequency	*HFE*	Metabolic	Hemochromatosis	Variable severity; organ damage if untreated.
	*CFTR*	Metabolic Pulmonary	Cystic fibrosis	Severe disease; early mortality without treatment.
	*BTD*	Metabolic	Biotinidase deficiency	Treatable; untreated may cause neurological symptoms.
	*GJB2*	Sensory (Hearing/Skin)	Vohwinkel syndrome/ Keratitis-ichthyosis-deafness	Non-fatal; significant sensory impact.
	*CYP21A2*	Endocrine	Congenital adrenal hyperplasia	Potential neonatal mortality without therapy.
	*GALT*	Metabolic	Classical galactosemia	Lethal in neonatal form if untreated.
	*SERPINA1*	Liver Lung	Alpha-1 antitrypsin deficiency	Early emphysema or liver failure; variable course.
	*PAH*	Metabolic	Phenylketonuria	Non-fatal with treatment; untreated causes severe disability.
	*SMN1*	Neuromuscular	Spinal muscular atrophy	Infantile forms fatal; treatable with gene therapy.
	*ATB7B*	Metabolic	Wilson’s disease	Fatal disease if not diagnosed and treated (copper accumulation in liver/brain).
	*USH2A*	Sensory (Vision/Hearing)	Usher syndrome type II	Non-fatal; dual sensory impairment.
	*WNT10A*	Craniodental Skin	Ectodermal dysplasia	Non-lethal; impacts quality of life.
Moderate frequency	*ACADM*	Metabolic	Medium-chain acyl-CoA dehydrogenase deficiency	Infant mortality risk if undiagnosed.
	*ALDOB*	Metabolic	Hereditary fructose intolerance	Severe hypoglycemia in infancy if untreated; treatable by dietary restriction.
	*DHCR7*	Metabolic Developmental	Smith–Lemli–Opitz syndrome	Lethal in severe forms; survivable with cholesterol supplementation.
	*GAA*	Metabolic Neuromuscular	Pompe disease	Infantile form lethal; treatable.
	*HBA1*	Hematologic	Alpha-thalassemia	Hydrops fetalis is lethal; trait forms are mild.
	*EVC*	Skeletal Growth	Ellis–van Creveld syndrome	Neonatal lethal forms.
	*SLC26A2*	Skeletal Growth	Diastrophic dysplasia	Severe skeletal dysplasia; perinatal lethal variants exist.
	*TPP1*	Neurodegenerative	Neuronal ceroid lipofuscinosis type 2	Early-onset neurodegeneration; fatal in childhood.
	*COL7A1*	Skin Connective tissue	Dystrophic epidermolysis bullosa	Severe forms fatal in childhood.
	*CYP11B2*	Endocrine	Aldosterone synthase deficiency	Can cause neonatal salt-wasting; treatable.
	*GBA1*	Metabolic Lysosomal	Gaucher disease	Infantile form lethal; chronic forms manageable.
	*NEB*	Neuromuscular	Nemaline myopathy	Severe neonatal forms fatal; variable severity.
	*NR2E3*	Sensory (Vision)	Neural ceroid lipofuscinosis	Non-lethal; causes visual impairment.
Low frequency	*ACAD9*	Metabolic Mitochondrial	*ACAD9* deficiency	Variable severity; can cause cardiomyopathy and early death if untreated.
	*BBS1*	Multisystem Developmental	Bardet–Biedl syndrome	Non-lethal; multisystem disorder affecting vision, obesity, and kidneys.
	*CAPN3*	Neuromuscular	Limb-girdle muscular dystrophy type 2A	Progressive; may shorten lifespan.
	*GALC*	Neurodegenerative	Krabbe disease	Infantile form is fatal.
	*SLC12A3*	Renal Electrolyte	Gitelman syndrome	Non-lethal; chronic electrolyte imbalance manageable with therapy.
	*SLC22A5*	Metabolic	Primary carnitine deficiency	Potentially fatal cardiac involvement if untreated; treatable with supplementation.
	*ARSA*	Neurodegenerative	Metachromatic leukodystrophy	Lethal infantile form.
	*CPT2*	Metabolic Neuromuscular	CPT II deficiency	Neonatal form lethal; adult form benign.
	*CRB1*	Sensory (Vision)	Retinal dystrophy Leber congenital amaurosis	Non-fatal; severe vision loss early in life.
	*EYS*	Sensory (Vision)	Retinitis pigmentosa	Non-fatal; progressive blindness.
	*G6PD*	Metabolic Hematologic	*G6PD* deficiency	Hemolytic anemia; rarely fatal if managed.
	*GBE1*	Metabolic	Glycogen storage disease type IV (Andersen disease)	Hepatic and neuromuscular forms; infantile form is often fatal.
	*HEXA*	Neurodegenerative	Tay–Sachs disease	Fatal in childhood.
	*LAMA2*	Neuromuscular	*LAMA2* muscular dystrophy	Congenital forms severe and lethal.
	*LDLR*	Metabolic Cardiovascular	Familial hypercholesterolemia	Premature cardiovascular disease; treatable with statins.
	*LIPA*	Metabolic	Lysosomal acid lipase deficiency	Wolman disease lethal in infancy.
	*MEFV*	Inflammatory Autoinflammatory	Familial Mediterranean fever	Non-fatal with treatment; risk of amyloidosis if untreated.
	*NPC1*	Neurodegenerative	Niemann–Pick type C	Infantile forms fatal; variable course.
	*VPS13B*	Developmental Neurological	Cohen syndrome	Non-fatal; developmental delay and visual impairment.

## Data Availability

Data are contained within the article and in the Appendix A.

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
