# Peer review of "The Landscape of Genetic Variation and Disease Risk in Romania: A Single-Center Study of Autosomal Recessive Carrier Frequencies and Molecular Variants"

_ijms, 2025, doi:10.3390/ijms262210912_

Round 1
Reviewer 1 Report
Comments and Suggestions for Authors
The study examined regional characteristics of the spectrum and frequency of heterozygous carriage of variants in the genes of autosomal recessive diseases within the framework of expanded carrier screening. This research topic is relevant, but the interest in the scientific and practical results of the work is local and associated with the Western Romanian region. The methods and approaches are adequate for this type of research. There are no major comments.
Minor comments:
- The abstract of the article can be reworked and the results can include primarily data on regional differences in the spectrum and frequency of variants;
- It is not entirely clear from the article from which sources the carrier frequencies presented in Table 1 in the Pan-ethnic carrier frequency column were taken.
Author Response
Thank you very much for taking the time to review this manuscript. Please find the detailed responses below and the corresponding revisions/corrections writen in red in the re-submitted manuscript.
Comment 1: The abstract of the article can be reworked and the results can include primarily data on regional differences in the spectrum and frequency of variants;
Response 1: Thank you for this valuable observation. We have revised and improved the abstract by adding more relevant data from the study. However, it is important to note that this is the first carrier screening study conducted in Romania, focusing on autosomal recessive and X-linked conditions. Therefore, we currently do not have a national cohort available for comparison. At the same time, we aim to remain cautious about generalizing our findings to the entire country. For this reason, we have consistently specified throughout the manuscript that our results represent data from a regional cohort in Western Romania.
Comment 2: It is not entirely clear from the article from which sources the carrier frequencies presented in Table 1 in the Pan-ethnic carrier frequency column were taken.
Response 2: Thank you for this comment. Since all patients included in our study were tested at Invitae Laboratories, the carrier frequencies listed in the Pan-ethnic column of Table 1 were extracted directly from the original laboratory reports. To clarify this point, we have now added an explanatory note in the Methods section. Additionally, we have attached an anonymized example of a laboratory report, where the pan-ethnic carrier frequencies mentioned in Table 1 can be observed.
Reviewer 2 Report
Comments and Suggestions for Authors
Miruna Gug and coworkers submitted the ms: The Landscape of Genetic Variation and Disease Risk in Romania: Comprehensive Analysis of Autosomal Recessive Carrier Frequencies and Molecular Variants
Comments for Authors
TITLE
The fact that data are from single center should be included in title; modify accordingly.
Introduction: in general OK;
Line 71:
71 “….may still transmit pathogenic alleles to their children.” Consider changing in: “they do have a 50% chance of transmitting each pathogenic heterozygous allele to their children”
Line 84:
“classify the observed variants to inform future ….” Consider changing in: “classify the observed variants, based on their frequency, to inform….”
Results
2.5 Classification of main morbidity risks
In my opinion the classification on morbidity risk is too simple, the clinical outcome of the quoted disorders may vary greatly according also to type of mutation (see for example infantile or adult type of Pompe disease: a carrier of a mutation may have children affected by infantile or adult form of the disease according to the mutation present in the partner); age of onset is also a parameter that should be taken into consideration.
My suggestion: remove this classification, and simply stress in the text with few examples the wide clinical variability that can be observed, and the relevance of the mutation found in the partner; or just separate disease with more restricted clinical variability.
In addition, section 2.6 and table 5 describes much more properly the issue previously discussed.
Lines 208 to 213 sound as a repetition, consider eliminating them.
Discussion
It should be rewritten: many data should be moved to results, there a lot of repeated data; certainly, it can be largely shortened. Obvious concepts as for instance those about prenatal diagnosis can be said with just a few words.
Author Response
Thank you very much for taking the time to review this manuscript. Please find the detailed responses below and the corresponding revisions/corrections writen in red in the re-submitted manuscript.
Comment 1: TITLE – The fact that data are from a single center should be included in the title; modify accordingly.
Response 1: Thank you for this helpful suggestion. We have revised the title accordingly to clearly indicate that the study is based on data obtained from a single medical center. This adjustment ensures that the scope and limitations of our dataset are transparent from the outset.
Comment 2: Introduction: in general OK.
Line 71: “….may still transmit pathogenic alleles to their children.” Consider changing to: “they do have a 50% chance of transmitting each pathogenic heterozygous allele to their children.”
Line 84: “classify the observed variants to inform future ….” Consider changing to: “classify the observed variants, based on their frequency, to inform….”
Response 2: Thank you very much for these precise and constructive observations. We have implemented both suggested changes in the Introduction section to improve accuracy and clarity. The revised sentences now read as recommended.
Comment 3: Results – Section 2.5: Classification of main morbidity risks
In my opinion, the classification on morbidity risk is too simple; the clinical outcome of the quoted disorders may vary greatly according to the type of mutation (for example, infantile or adult type of Pompe disease). Age of onset should also be taken into consideration.
My suggestion: remove this classification, and simply stress in the text with a few examples the wide clinical variability that can be observed, and the relevance of the mutation found in the partner; or just separate diseases with more restricted clinical variability.
In addition, section 2.6 and Table 5 describe much more properly the issue previously discussed.
Lines 208–213 sound repetitive; consider eliminating them.
Response 3: Thank you for these valuable and detailed recommendations. We have carefully considered all your suggestions and decided to merge Sections 2.5 and 2.6 to streamline the presentation of results. The simplified structure now highlights examples illustrating clinical variability and the importance of mutation type and partner genotype, as suggested. We have also removed redundant sentences (Lines 208–213) to avoid repetition and improve readability.
Comment 4: Discussion
It should be rewritten: many data should be moved to Results; there are a lot of repeated data. Certainly, it can be largely shortened. Obvious concepts, such as those about prenatal diagnosis, can be stated with just a few words.
Response 4: We appreciate your insightful feedback regarding the Discussion section. Following your recommendations, we have thoroughly revised and condensed this section. Several data descriptions have been moved to the Results section to avoid redundancy. The rewritten Discussion is now more concise, focused, and emphasizes key findings in a clearer and more engaging manner for the reader.
Round 2
Reviewer 2 Report
Comments and Suggestions for Authors
Dear Author,
thanks for having taken in consideration my suggestions.
I feel the ms was greatly improved.
Author Response
Dear Reviewer,
Thank you again for all your valuable suggestions, which have greatly helped us to improve and clarify the manuscript. We sincerely appreciate your time and constructive feedback throughout the review process.
Best regards from our team!